IMU-aided adaptive mesh-grid based video motion deblurring

Arslan Ahmet 1
Gultekin Gokhan Koray gkgultekin@aybu.edu.tr 2
Saranli Afsar 3
1 Center for Image Analysis (OGAM), Middle East Technical University , Ankara , Turkey
2 Electrical and Electronics Engineering, Ankara Yildirim Beyazit University , Ankara , Turkey
3 Electrical and Electronics Engineering, Middle East Technical University , Ankara , Turkey
Aleem Muhammad
Electronic publication date: 2024 Nov 25
Publication date: 2024
Volume: 10
Electronic Location ID: e2540
Received 2024 Aug 28; Accepted 2024 Nov 4
Copyright: ©2024 Arslan et al.
Copyright year: 2024
Copyright holder: Arslan et al.
License: This is an open access article distributed under the terms of the Creative Commons Attribution License, which permits unrestricted use, distribution, reproduction and adaptation in any medium and for any purpose provided that it is properly attributed. For attribution, the original author(s), title, publication source (PeerJ Computer Science) and either DOI or URL of the article must be cited.
License URL: https://creativecommons.org/licenses/by/4.0/

Keywords: Cameras, Non-uniform motion blur, Motion deblurring, Inertial measurement unit, Blur kernel, Image restoration, Point spread function

Funding: ROKETSAN This research is funded by ROKETSAN. The funders had no role in study design, data collection and analysis, decision to publish, or preparation of the manuscript.

==============================
Motion blur is a problem that degrades the visual quality of images for human perception and also challenges computer vision tasks. While existing studies mostly focus on deblurring algorithms to remove uniform blur due to their computational efficiency, such approaches fail when faced with non-uniform blur. In this study, we propose a novel algorithm for motion deblurring that utilizes an adaptive mesh-grid approach to manage non-uniform motion blur with a focus on reducing the computational cost. The proposed method divides the image into a mesh-grid and estimates the blur point spread function (PSF) using an inertial sensor. For each video frame, the size of the grid cells is determined adaptively according to the in-frame spatial variance of blur magnitude which is a proposed metric for the blur non-uniformity in the video frame. The adaptive mesh-size takes smaller values for higher variances, increasing the spatial accuracy of the PSF estimation. Two versions of the adaptive mesh-size algorithm are studied, optimized for either best quality or balanced performance and computation cost. Also, a trade-off parameter is defined for changing the mesh-size according to application requirements. The experiments, using real-life motion data combined with simulated motion blur demonstrate that the proposed adaptive mesh-size algorithm can achieve 5% increase in PSNR quality gain together with a 19% decrease in computation time on the average when compared to the constant mesh-size method.

Introduction

The cameras on mobile platforms capture image sequences that are used for many purposes. Often, mobile platforms fail to create steady conditions for image capture. One of the difficulties is motion blur due to camera translation and rotation during the frame exposure time. Motion blur degrades the visual quality of the image, results in information loss, and hence affects the performance of image processing algorithms (Gultekin & Saranli, 2019a). This effect is most pronounced for fast-moving mobile platforms and has a significant impact on many relevant machine-vision applications such as simultaneous localization and mapping (SLAM) and visual odometry (Qu et al., 2024; Wang & Huang, 2014). It also results in the degraded quality of video recordings from platforms such as uncrewed aerial vehicles (UAVs) (Oktay, Celik & Turkmen, 2018).

When translation and rotation of the camera are present, the rotation is often the dominant effect in motion-blur, in particular when the focal length is small and/or the object plane is reasonably far. In order to remove motion blur from a blurred video frame, the point spread function (PSF) (also known as blur kernel) should be estimated and an inverse-filtering (deconvolution) operation needs to be performed to obtain the restored image frame (Gultekin & Saranli, 2019b). This is an ill-posed problem because motion-blur is a lossy process, i.e, some information in the frame is lost. In the literature, motion deblurring algorithms are generally classified into two categories based on the available prior information about the PSF. Blind deconvolution algorithms assume that the PSF isn’t available and should also be estimated from the same frame prior to deconvolution. Blind algorithms are generally much slower with lower performance, especially under large and spatially-variant blur. There are however some recent developments in neural networks and deep-learning (Kupyn et al., 2019; Zamir et al., 2021; Zhang et al., 2019) where improved blind deblurring quality is reported. Unfortunately, high computational cost and requirement for specialized hardware such as GPUs is still a drawback.

In non-blind deconvolution, an external source such as an inertial measurement unit (IMU) composed of a three-axis gyroscope and three-axis accelerometer can be used to obtain information about the camera motion. By considering the intrinsic and extrinsic camera calibration parameters and the projection into the image plane, the resulting PSF can be estimated (Gültekin & Saranlı, 2017). In both cases, a chosen deconvolution technique can be used to recover the deblurred image by using the estimated PSF. In the non-blind case, since the PSF is estimated quickly without heavy computation and with better precision, deblurring is also faster and performance in terms of a deviation from the original non-blurred sharp image is improved. The use of external sensors can efficiently measure 3D camera motion but the projection into the image plane still requires additional assumptions such as knowledge of the scene depth across the image frame (Boyacıet al., 2014). Hu et al. (2016), Joshi et al. (2010), and Mustaniemi et al. (2018) use IMU data with non-blind deconvolution techniques to achieve effective deblurring. However, relevant issues such as spatial non-uniformity of the blur, especially due to camera roll motion as well as computation time remains as challenges.

In this paper, we present a non-blind deblurring algorithm that divides the blurred image into a mesh-grid and employs IMU measurements to estimate non-linear and non-uniform PSFs for each grid cell. We further improve and extend the mesh-division approach with the introduction of an adaptive mesh-size that is determined dynamically using the spatial variance of the camera motion vectors across each frame. Two variants are initially studied, namely one that determines the mesh-size to achieve the highest quality and the other for a balance between quality and computational cost. These ideas are combined in a trade-off parameter that allows the user to explore the algorithm performance/complexity tradeoff on a two-dimensional performance plane. The experimental results, conducted on physical motion data and real images but combined with simulated motion blur show that our novel approach can increase the deblurring quality with decreasing computational cost. Our results are visualized in Fig. 1. The contributions of this study are outlined as follows:

Figure 1 The blurred and deblurred frame examples from the datasets.

The blurred frames are deblurred with different mesh-sizes obtained from the HQ and SQB algorithms along with classical approach (m:480). The values of variance, mesh-size, peak signal to noise ratio (PSNR) and proccesing times are given below each frame.

• A novel adaptive mesh-size determination method is proposed that dynamically adjusts the mesh size based on the spatial variance of camera motion vectors, enhancing the mesh-division approach.

• Two algorithmic variants are studied: One variant prioritizes achieving the highest deblurring quality. Another variant balances between deblurring quality and computational efficiency.

• A quality/speed trade-off control parameter is introduced, allowing users to control the adaptive mesh-size determination process according to the application’s quality/speed requirements.

The paper begins in ‘Related Work’ by a review of the relevant study in the literature. We then present the background theory in ‘Background’ followed by the proposed adaptive mesh-grid solution to effectively handle non-uniform and non-linear blur in ‘Proposed Method for Addressing Non-uniform Motion Blur’. Our experimental results and the proposed trade-off parameter is presented and discussed in ‘Datasets’. The paper concludes in ‘Conclusion’ with our main conclusions.

Related Work

Motion blur is often represented as a convolution problem. The restoration process encompasses firstly the estimation of the blur kernal function or PSF. This is followed by a “deconvolution” operation to restore the latent image. The deconvolution from a single image is an ill-posed problem since the blur filter loses information in some frequencies as a function of the magnitude of the blur (Gultekin & Saranli, 2019a). The literature contains numerous blind and non-blind deblurring algorithms designed to recover the latent sharp image. In addition to the well-established deblurring algorithms, there are more recent attempts with increasing usage of neural networks (Zhang et al., 2022) and reinforcement-based deep learning methods (Kupyn et al., 2019; Zamir et al., 2021).

Blind deblurring algorithms aim to simultaneously estimate the PSF and the latent image using only the provided image data. Most algorithms approach the problem by setting up a minimization framework and applying regularizers or prior information about the image statistics to cope with the missing (lost) information (Pan et al., 2016; Krishnan, Tay & Fergus, 2011; Perrone & Favaro, 2014; Xu, Zheng & Jia, 2013; Fergus et al., 2006; Cai et al., 2009; Levin et al., 2011). Some modeling efforts for natural image statistics in the literature include for example the Gaussian image prior (Fergus et al., 2006), L0-regularized prior (Xu, Zheng & Jia, 2013; Shan, Jia & Agarwala, 2008), dark channel prior (Pan et al., 2016), and extreme channel prior (ECP) (Khongkraphan, Phonon & Nuiphom, 2021) which combines dark channel and bright channel prior. Methods that rely on image priors usually manage slight noise and outliers effectively. However, their performance notably declines when dealing with blurred images containing saturated pixels and moderate levels of noise (Dong et al., 2017). In addition, the computational cost of blind deconvolution algorithms is in general very high.

Non-blind deblurring methods operate under the assumption that the PSF is provided as an input. When the blur is due to the camera ego-motion, this PSF can itself be estimated using for example measurements from inertial measurement units (IMUs). Recently, these sensors have become widely available as part of or in association with most cameras, e.g., on smartphones. There exist considerable prior work on inertial-aided non-blind image deblurring algorithms that use an IMU in order to estimate the PSF (Varghese, Rajagopalan & Ansari, 2024; Mustaniemi et al., 2018; Sindelar & Sroubek, 2013; Joshi et al., 2010; Hu et al., 2016; Park & Levoy, 2014). These studies utilize motion measurements from gyroscopes and/or accelerometers collected during the camera’s exposure time to project the effect of motion onto the image plane and estimate the resulting blur kernel. Directly estimating the PSF in this manner offers a substantial computational advantage over blind algorithms, making these methods suitable for real-time applications.

Before further discussing IMU aided non-blind deblurring algorithms which are our main focus, one should also mention some recent examples of neural network and deep learning-based blind and non-blind image deblurring methods (Kim et al., 2024; Zhang et al., 2022; Kupyn et al., 2019; Zamir et al., 2021; Zhang, Tang & Qu, 2024; Asim, Shamshad & Ahmed, 2020; Tran et al., 2021; Mustaniemi et al., 2019). In study Kupyn et al. (2019), a generative adversarial network (GAN) is proposed in combination with a new feature pyramid network as a further improvement of a previous study (Kupyn et al., 2018). In Zamir et al. (2021), multi-scale and scale-recurrent architectures are employed with a multi-level convolutional neural network (CNN) model to reduce computational costs and tackle quality issues associated with deep-learning-based methods. A non-blind, inertial measurement-based CNN network is suggested in Mustaniemi et al. (2019). This study is the first to effectively combine inertial measurements with deep learning networks for deblurring. Despite recent advancements in this field, significant issues persist, including the need for powerful and often costly GPU systems, performance that depends on the dataset, and lengthy training times.

When the PSF is known, non-blind deblurring techniques can be used; however, this approach does not resolve the underlying ill-posed nature of the deblurring problem. To address the ill-posed nature of the deblurring problem, these technique also utilize natural image statistics and set up the solution as an optimization problem. Various approaches have been developed to integrate non-blind deblurring techniques with natural image statistics (Levin et al., 2007; Fortunato & Oliveira, 2014; Wang et al., 2008; Zhou & Nayar, 2009). Gaussian and hyper-Laplacian priors are used in Levin et al. (2007). Using Gaussian priors leads to a linear system that simplifies the solution process, but can also cause ringing artifacts. Yuan et al. (2008) introduced a multi-scale method that incorporates residual deconvolution at each scale. On the other hand, hyper-Laplacian priors involve solving a more complex non-convex optimization problem. To address this, Krishnan & Fergus (2009) introduced a solution method using half-quadratic minimization (HQM), which is also a component of the total variation method (TV- L1) in Wang et al. (2008).

Another observation on the existing literature is that most of the inertial-aided algorithms (Joshi et al., 2010; Hu et al., 2016; Park & Levoy, 2014; Sindelar & Sroubek, 2013) assume spatially-invariant (or uniform) blur in the image. When there is rotational camera motion around the optical axis, this assumption is invalid, leading to a significant degradation in deblurring quality. To handle this spatially-varying blur, Mustaniemi et al. (2018) developed and applied the mesh-grid decomposition technique. The image is divided into a constant-grid-size rectangular mesh and now, grid-cell specific PSFs are estimated and assumed spatially-invariant (uniform) within the cell. A similar mesh-grid based approach is used in Hirsch et al. (2011) with a blind PSF estimation.

Considering the much-degraded performance of IMU-assisted non-blind deblurring methods under spatially-varying (non-uniform) blur, we mainly focus on the process of IMU-assisted PSF estimation and examine how spatial variance affects deblurring. Our study builds upon the mesh-grid method introduced in Mustaniemi et al. (2018) but improves it substantially for the case of an image video sequence mainly taking into account the temporal variation of the spatial non-uniformity. The mesh-grid approach involves partitioning the blurred image into grid cells, estimating the PSF for each cell using the inertial measurements, and subsequently applying non-blind deconvolution to each cell. The mesh cell-size is not fixed to a predetermined value but adaptively changes for each frame according to a proposed, easily computable blur non-uniformity metric within that frame.

Background

The determination of the motion-blur PSF because of the camera motion on each pixel of the image plane requires the assumption of a static scene (no scene motion) and the projection of the camera motion onto the image plane (Mutlu, Saranli & Saranli, 2014). In this section, we describe this projection based on the inertial measurements of the camera motion. The obtained blur PSF is later used for the non-blind deblurring(deconvolution) process.

Camera model and projection of motion

Given the 6D camera movement in space, the well known pinhole camera model is used to calculate the apparent motion of an object plane point on the image plane (Mutlu, Saranli & Saranli, 2014). In Fig. 2, the image plane is oriented parallel to the x − y plane and is located a distance f away from the origin along the z-axis, which represents the optical axis.

Figure 2 Definition of coordinate axes for the camera projection model.

The z axis represents the principal (optical) axis, and f denotes the camera’s focal length.

Rotations around the x, y and z (optical) axis are defined as roll pitch and yaw respectively. World (object plane) coordinates (X, Y, Z) are projected onto the image plane with coordinates (x, y) using the pinhole camera model, (1) xy=fXZfYZT,

where f is the camera focal length. By differentiating Eq. (1) with respect to time, we obtain the velocities (u, v) of image points caused by the camera’s motion relative to a stationary world point. (2) uv=fX ˙Z−XZ ˙Z2Y ˙Z−YZ ˙Z2=fZ0−fXZ20fZ−fYZ2X ˙Y ˙Z ˙.

Assuming rotational motion of the camera around the three-axis, and if the camera rotational velocities are given by wx, wy and wz we have the apparent linear velocities given by (3) X ˙Y ˙Z ˙=0−ωZωYωZ0−ωX−ωYωX0XYZ.

Substituting Eqs. (3) into Eq. (2) and also using Eq. (1) gives us: (4) uv=ωzy−ωyf−ωyfx2+ωxfxyωzx−ωxf−ωyfxy+ωxfy2.

Equation (4) calculates image plane velocities resulting from the camera rotational motion. If the relative translational velocities are also considered, the full effect of the camera motion on the image plane pixel velocities becomes: (5) uv=ωzy−ωyf−ωyfx2+ωxfxy−fzvx+xzvzωzx−ωxf−ωyfxy+ωxfy2−fzvy+yzvz.

Knowledge of the image plane pixel velocity vectors using Eq. (5) allows us to obtain the motion blur PSF. As can be seen from this expression, the effect of the translational camera motion is inversely proportional to camera-object distance z. When this distance is large compared to the camera focal length f, the contribution from the translational motion can be neglected. The effect of the rotational motion is independent of the distance. For the present study, we assume a reasonable distance between the object plane and the camera and hence neglect the translational motion and use Eq. (4) in our PSF estimation.

Inertial measurements

The rotational motion data is acquired through inertial measurements. In Fig. 3, the motion of the camera within the integration time is illustrated together with the corresponding projection onto the image plane. Within the exposure time, the world point (X, Y, Z) is mapped to various locations on the image plane due to the camera motion. Assuming that an inertial measurement is available at each camera location, the corresponding pixel velocity vector on the image plane can be calculated using the Eq. (4). Hence, the projection of the world point on the image plane at each sample time during the exposure is given by:

Figure 3 Camera motion trajectory during exposure time.

K samples of camera motion are captured from IMU measurements and corresponding image plane motion trajectory is calculated.

(6) xk,yk=x0,y0+ ∑j=1kuj,vj∗Δt

for k = 1, …, K. Here, Δt is the IMU sampling period and (uj, vj) is the projected 2D pixel velocity for the j th IMU measurement. Equation (6) gives us the complete image plane motion trajectory for the estimation of the blur PSF.

PSF estimation

(7) hm,n=1K∑k=1Kδm−xk,n−yk.

The accuracy of this estimated motion trajectory is limited by the accuracy of the IMU measurements. The inertial measurements, which are the gyroscope measured turn rates in our case are corrupted by noise. Noisy measurements cause erroneous PSF estimation and this leads to poor deblurring results.

Considering the presence of noisy measurements, we make a Gaussian distribution assumption for projected points on the image plane over the space variables as proposed in Lee et al. (2014). The parameters of this Gaussian distribution (mean and variance) are estimated experimentally using inertial measurements captured from a stationary IMU device when no camera motion is present. A Monte-Carlo method is used, namely the collected 3-axis turn rate samples are each transformed into image planes (u,v) samples. These samples are used to estimate the mean and variance of the image plane velocity distribution. Figure 4 illustrates the resulting estimates. The distributions of single points are illustrated at the bottom part of Fig. 4. The bottom right image depicts the ideal single point PSF, whereas the left image represents the PSF with a Gaussian distribution.

Figure 4 (Upper left) The histogram of u and v axis velocities generated from the stationary measurements. (Upper right) Estimated PSF using inertial data: Corresponding non-linear blur PSF is estimated from the projected motion trajectory (Eq. 4) between the start point and end point using the sequence of image plane velocity vectors and Eq. (8). (Bottom) A Gaussian distribution is fitted to this data to model the effect of the noise on the PSF estimation.

The 3x3 masks show a single PSF point produced by this Gaussian distribution (left) and the ideal noise-free PSF (right).

Combining the PSF equation in Eq. (7) with this uncertainty due to the measurement noise, the resulted PSF in the image plane can this time be formulated as the sum of sample Gaussian distributions as: (8) hm,n=1KG ∑k=1KGm−xk,n−yk,

where G is a two-dimensional Gaussian distribution whose variance is determined with the aforementioned method and whose arguments are the corresponding Gaussian means along u and v axes. KG is a normalization constant for preserving the image intensity.

In Fig. 4, the image plane motion during the integration period as well as the fully estimated noisy PSF is illustrated. Each distinct color (linear segment) in the trajectory corresponds to a distinct motion vector obtained from inertial measurements. The estimated noisy PSF (top right) is obtained from the trajectory by using the noisy PSF model obtained in Eq. (8) as a sum of shifted Gaussians. This PSF estimate is used for the deconvolution-based deblurring of the corresponding frame. Note that the intensity value of a pixel in the PSF model corresponds to the time the projected object point spends on the image plane on that point.

Non-uniform blur, as the main focus of our study, is a spatial variation of the blur PSF within an individual frame. For the case where the object plane is at a distance and parallel to the image plane, only the roll motion of the camera results in a non-uniform blur. The estimation of a single PSF at a particular point of the image plane was explained in the previous sections. The process of handling non-uniform PSF within a frame is explained in this section. Within the considered mesh-grid framework, multiple PSFs, each associated with a particular mesh grid-cell are used.

Proposed Method for Addressing Non-uniform Motion Blur

One group of existing deblurring algorithms typically assume uniform motion blur and use a single PSF for the entire image. However, camera motion and roll motion in particular necessarily result in non-uniform blur. Assuming uniform blur in this case severely impacts the performance of deblurring algorithms.

An example image plane motion vector field that yields nonlinear and non-uniform motion blur in our data set is shown in Fig. 5. Two red boxes are selected as example regions of the image where the motion trajectories are very different. Enlarged views of the trajectories and resultant PSFs are shown at the bottom of the image. The PSFs over the frame vary significantly which indicates the level of non-uniformity within the image frame. In the classical approach of non-blind methods, there would be a single averaged PSF for all parts of the image frame. However, this PSF will only be accurate for a specific region of the image, causing substantial estimation errors in other areas. To effectively handle non-uniform blur and apply it to deblurring of real-time video, we consider the mesh-grid decomposition idea from the literature and then extend it by introducing an adaptive mesh-size concept for accurate and fast PSF estimation for a sequence of video frames.

Figure 5 (Top) Illustration of the non-uniform blur distribution over the image plane. Motion trajectories on the image plane are created from 10 IMU measurement samples within the frame integration time. (Bottom) Two selected blur PSFs are enlarged to show the significant difference between them.

Mesh-grid decomposition for non-uniform blur

Mesh-grid based deblurring technique is proposed to deal with the non-uniformity of motion blur (Mustaniemi et al., 2018). The mesh-grid decomposition of an image frame is shown in Fig. 6. In our study, the grid-cell size is not constant and an overlap region between grid-cells is also defined as will be explained in the subsequent discussion.

Figure 6 Dividing the image into a mesh grid and adding an overlap region is described.

Grid cells of size (m × m) pixels are extended by an overlap size of n pixels. Using overlap regions reduces the ringing artifacts at the borders of grid cells during deconvolution and is removed during the merging process.

The frame is split into meshes denoted as Mi,j where i is the number of rows and j is the number of columns of grid-cells. Equation (4) is used to determine the motion vectors on image plane from the rotational camera motion. The motion vectors and the corresponding PSFs for each grid-cell is calculated according to its center pixel using Eq. (8). PSF of a cell is denoted as Hi,j, analogous to mesh grid-cells.

Deblurring process causes some artifacts at the borders of the image which is known as ringing, especially for large PSF kernel sizes. In mesh-grid based deblurring, these ringing artifacts appear at the borders of each cell since each grid-cell is deblurred like an individual image. Therefore, mesh-grid seams appear in the deblurred image when the deblurred grid cells are combined together. We observed that without any overlap region these artifacts are highly visible. Therefore, differing from the previous method (Mustaniemi et al., 2018), we use overlap regions at the mesh-grid cell structure and edge-tapering to decrease the ringing artifacts and minimize the visibility of mesh-grid seams in the deblurred image. The overlap regions are removed once the deblurring process of the cells is completed.

Figure 7 illustrates the combined effect of edge tapering and the overlap region used with the mesh-grid structure. It is observed that artifacts are highly visible when neither method is used. Using edge tapering alone exposes the seams of the mesh grid and the tapering effect at the borders of each grid. Based on these observations, we chose to apply both the edge tapering and overlap region. The overlap size (n) is set to the maximum blur kernel size (Hmax). (9) n=Hmax= maxsizeHi,j

Figure 7 The image deblurring results are presented for both cases: with and without overlap regions and edge tapering.

The width of the overlap region varies from 0 pixels to the maximum dimensions of the PSF. Applying edge tapering along with an overlap size of Hmax results in the highest quality.

Here, i and j represent the mesh-grid indices for rows and columns, respectively. Like edge tapering, the overlap size is adjusted based on the blur magnitude. Increasing the overlap size beyond a certain point does not improve the results. Therefore, adding an overlap region equal to the maximum size of PSFs yields the optimal result in terms of minimizing both mesh-grid and ringing artifacts.

Proposed solution of adaptive mesh-size

Using a constant mesh-size for all frames is not efficient since motion blur is mostly different in each frame. Therefore, it is important to determine a suitable value for each frame that yields a reasonable result in terms of both accuracy and computation speed.

The core concept of the adaptive mesh-size algorithm is to estimate the blur non-uniformity in the image frame and adjust the mesh size accordingly. We estimate this non-uniformity using the variance of motion vector magnitudes, which allows us to effectively manage spatially-varying blur across a sequence of frames. This approach is illustrated in Fig. 8.

Figure 8 Illustration of adaptive mesh size: as blur non-uniformity decreases, the calculated frame variance also decreases.

This allows selecting larger grid cell sizes and reducing computational cost.

The peak signal-to-noise ratio (PSNR) is used as a quality metric in this study which is a widely used metric in image processing, particularly in deblurring tasks to measure the quality of reconstructed or processed images in comparison to a reference (ground truth) image. It quantifies how much the restored image differs from the original, making it an essential tool for evaluating deblurring accuracy. A higher PSNR value indicates that the reconstructed image is more similar to the original image, implying better deblurring performance.

PSNR is calculated using the mean squared error (MSE) between the original image Ioriginal and the deblurred image Ideblurred as given in Eqs. (10) and (11). (10) PSNR=10⋅log10MAX2MSE

where MAX is the maximum possible pixel value of the image and MSE is calculated using Eq. (11). (11) MSE=1MN∑i=1M ∑j=1NIoriginali,j−Ideblurredi,j2

where M, N represent the dimensions of the image and i, j represent the indices of the pixels respectively.

Adaptive mesh-size for high-quality deblurring

The accuracy of the deblurring process is closely related to the PSF estimation error. The accuracy of the PSF can be improved by reducing the mesh-size to the smallest value that is possible. However, there are also limitations over the mesh-size due to blur magnitude, ringing artifacts and seam artifacts. In the preceding sections, we addressed the artifact issues and defined the overlap region. There are two constraints over the minimum size of grid cells: it must be larger than Hmax for deconvolution and it must be twice Hmax for edge tapering as stated in Krishnan & Fergus (2009). Our approach for the overlapping regions addresses these constraints since the overlap size is set to Hmax along every dimension, increasing the overall size of grid cells by twice Hmax.

This overlap region structure makes it possible to use very small mesh sizes. However, the main issue is that most of the computation time is spent on processing the overlap region. For instance, if we have a blur magnitude of 30 pixels and select a [2 × 2] pixel mesh size, the overlap region would be 30 pixels in each dimension. Thus, the total grid-cell size to be processed becomes [62 × 62] pixels, but only the [2 × 2] pixel region is used for the deblurred image and the rest is cropped out. Consequently, nearly the entire processing time is taken up by the overlap region., which is eventually discarded. Considering this, we concluded that the mesh size needs to be at least equal to the overlap size. This ensures that the total mesh size computation time is not significantly impacted by the overlap region.

Considering these constraints, the highest PSNR gain is achieved with the smallest possible mesh size, referred to as high quality mesh-size (mHQ) in Eq. (12). (12) mHQ=2∗Hmax=2∗ maxsizeHi,j

where i is the row and j is the column indexes of mesh-grid respectively.

Adaptive mesh-size for speed-quality balanced deblurring

The performance of the proposed HQ algorithm is satisfactory corresponding quality. However, the high computation cost can be a major concern in some applications. Therefore, we introduce another adaptive mesh-size approach to control the quality and speed trade-off. Figure 9 shows the results for PSNR gain and mesh size versus computation time for an image sample. As the mesh size decreases, processing time increases exponentially, while PSNR gain remains nearly constant with minimal improvement (purple box region). Our objective is to find a speed-quality balanced (SQB) point within the green box region, where a reasonable PSNR gain can be achieved without a substantial increase in computation time.

Figure 9 The graph displays processing time and PSNR gain as functions of mesh size for a sample image having a blur variance of 12 pixels.

The computational cost exponentially grows, while the PSNR gain improves more linearly. Choosing a mesh size within the green boxed area offers a reasonable balance between computation time and PSNR gain.

The speed-quality balanced adaptive mesh-size approach considers the non-uniformity of the blur. A larger mesh size is chosen when the differences in PSFs between regions are smaller, while a smaller mesh size is used when these differences are larger. The amount of non-uniformity of PSFs over the image is represented by a proposed blur magnitude spatial variance (σs2) parameter.

Mesh-size determination for speed-quality balanced deblurring

The relation between the PSF variance, computation time, and deblurring quality is experimentally determined by manual labeling the optimal points over Dataset-1. This dataset is explained in ‘Datasets’ later. Our goal is to create a variance versus mesh-size graph by manually selecting the results that balance speed and quality. We consider the area shown with a green box as an optimal region for this purpose. An exponential curve is fitted to this data which yields the equation for the mesh-size that balances speed and quality, based on the variance of PSF magnitudes.

A sample blurred image is deblurred using various mesh sizes, and the resulting images are presented in Fig. 10. This might offer a clearer understanding of the process for selecting optimal points. The difference in gain between mesh sizes of 96, 120, and 160 pixels is minimal. Significant decline in quality occurs at 240 pixels mesh-size from 2.23 dB to 1.13 dB. This can be considered the upper limit for the mesh size. The highest quality result is achieved with a 60-pixel mesh size, yielding a 2.82 dB PSNR gain. However, the computation time for this mesh size is 1.26 s, which is about 10 times slower than deblurring with a full-frame (480-pixel mesh size). When we examine the 96 and 120 pixels mesh-sizes, We observe a negligible difference in PSNR gains, amounting to just 0.07 dB.

Figure 10 Images deblurred using different mesh-sizes (M.size) are shown together with original and blurred images. Corresponding PSNR-Gain (P.Gain) in dB, and computation time(C.Time) in seconds are indicated on top of the images.

The choice of speed-quality balanced points was carried out according to computational efficiency, PSNR gain and visual quality.

Additionally, the conventional full-frame deblurring result (with a 480-pixel mesh size, shown in the bottom-right image of Fig. 10) exhibits a PSNR of −0.85 dB, which is the worst one across all results. This indicates that deblurring can sometimes degrade the quality of blurred images, if the PSF is estimated based on just the image center, which may not be accurate for all frames under roll motion of camera. This highlights the effectiveness of the mesh-division method in handling spatially varying blur.

In our approach, a manual optimal point is determined considering the PSNR-gain differences and computational time. The optimal points are determined for each frame of the synthetic dataset using this method. We applied this approach to select SQB mesh sizes for Dataset-1, and the results are shown in Fig. 11. The data is sorted to facilitate a clearer comparison of the effects of variance, mesh size, and processing time.

Figure 11 (Left) Total frame computation time and mesh-size results with respect to the variance for selected speed-quality balanced points from Dataset-1. (Right) Exponential model for the speed-quality balanced mesh size calculation based on blur variance.

The points were selected manually, and the mesh sizes are divisors of a 480 × 480 image.

Figure 11 presents the graph of variance versus speed-quality balanced mesh size, built according to two criteria: PSNR gain and computation time. Our HQ mesh-size algorithm converges to the smallest possible mesh size, producing the highest achievable deblurring quality. However, this leads to a significant increase in computation time, as shown in Fig. 11. In cases of high blur variance, smaller mesh sizes are effective for achieving a high quality deblurring result. However, when the blur variance is low, the HQ mesh size becomes inefficient because the spatial differences between estimated PSFs are minimal. In such cases, using a larger mesh size may cause a negligibly small decrease in PSNR gain but significantly reduces computation time. Therefore, the proposed variance parameter is shown to be a good indicator for the decision of a reasonable mesh-size. (13) mSQBσs2=422∗e−0.16∗σs2+87.25∗e−0.000824∗σs2

where σs2 is the spatial variance of blur magnitude, indicating the level of non-uniformity in the blurred frame. Additionally, the SQB mesh-size is constrained by the blur magnitude, which also limits mHQ. Consequently, the SQB mesh-size cannot be reduced under twice the blur magnitude, ensuring that mSQB is not less than mHQ.

Experiments and Test Results

Datasets

For the evaluation of the proposed methods and their performance, we generated four synthetic datasets using the IMU data and the videos published in sources (robetm101, 2019; TungArt7, 2024). Inertial data was captured from an MPU-6050 IMU device at a 500 Hz sampling rate during rapid walking. The camera parameters were set to 1.4 mm focal length and 3.39 µm sensor size. We assumed the camera to be ideal and perfectly aligned with the scene. The exposure time was set to 0.02 s, allowing for 10 IMU samples per frame.

We created four datasets, labeled Dataset-1 through Dataset-4, each serving different purposes and exhibiting various content and blur characteristics. Dataset-1 was utilized to experimentally determine the optimal SQB mesh size. We analyzed the deblurring results of this dataset and determined the best mesh sizes both for quality and speed. Datasets 2, 3, and 4 were used to evaluate our methods against traditional deblurring approaches. The generated datasets include non-uniform and non-linear motion blurred images.

The blur characteristics of the datasets are visualized using graphs in Fig. 12, together with histograms showing the average blur magnitude and its variance. Some sample blurred image frames are shown in Fig. 13. Dataset-1 consists of 100 frames, while other datasets consist of 50 frames, all with 480x480 pixels resolution. The magnitude of blur for every frame is calculated by determining the sizes of blur vectors based on the inertial data according to Eq. (4). This reveals the blur levels in the datasets. Additionally, the variance of the blur magnitudes offers insight into the non-uniformity of the blur within the images and across the frames in the dataset.

Figure 12 The inertial measurements are shown accompanying with the corresponding histogram of average blur magnitude and histogram of blur magnitude variances for every frame in Datasets 1, 2, 3 and 4 from top to bottom, respectively.

The blur magnitude is calculated by determining the maximum size of PSFs that are estimated.

Figure 13 Examples of original and blurred images from Datasets 1, 2, 3 and 4.

The primary difference between these datasets lies in the characteristics of the blur within the images. Dataset-2 contains more uniformly blurred images, as indicated by its lower variance. This suggests that the motion in Dataset-2 is more dominant in pitch and yaw rotations. Conversely, Dataset-3 exhibits significant roll motion around the optical axis, resulting in higher variance levels compared to Dataset-2. Dataset-4 features a more balanced motion, with similar magnitudes of blur across all three rotational axes, indicating a combination of pitch, yaw, and roll movements.

Comparative analysis of constant vs. adaptive mesh-size algorithms

In this section, we compare of our adaptive mesh-size algorithm and the available constant mesh-size method with respect to the deblurring performances. We claim that, selecting the mesh-size as a constant value is not effective in terms of quality versus computation time. For example, if the roll motion in a frame is small, selecting a small mesh-size increases computation time significantly without any notable increase in the deblurring quality.

Dataset-2, Dataset-3 and Dataset-4 are deblurred with different constant mesh-sizes and with our adaptive HQ and SQB algorithms. The performance comparison is given in Table 1 and some visual results are shown in Fig. 1. It is observed that SQB algorithm achieves slightly better PSNR gain with a lower computational cost (e.g., the cases shown in bold in Eq. (1). The HQ algorithm, achieves slightly better PSNR gain among all methods. However, its computation time is approximately 10 times higher compared to the SQB method.

Table 1 Comparison of SQB and HQ algorithms with the constant mesh-size method.

The average PSNR gain and computation time results on Dataset-2, Dataset-3 and Dataset-4 are presented. Bold values illustrate SQB algorithm achieving a superior PSNR gain/computation time balance when compared to the nearest constant mesh size case (Last row shows the percent increase in PSNR gain and percent decrease in computation time achieved by SQB). Entries marked with an asterisk (*) denote instances where the blur magnitude exceeded the mesh size, resulting in diminished PSNR performance.

	Dataset-2	Dataset-3	Dataset-4	
	Avg.	Avg.	Avg.	
Mesh-Size	PSNR gain (dB)	Time (s)	PSNR gain (dB)	Time (s)	PSNR gain (dB)	Time (s)	
480 × 480	1.34	0.15	−0.91	0.15	0.01	0.14	
240 × 240	2.50	0.25	1.06	0.28	2.05	0.28	
160 × 160	3.01	0.40	1.89	0.45	2.47	0.43	
120 × 120	3.31	0.62	2.27	0.67	2.67	0.66	
96 × 96	3.48	0.96	2.47	0.98	2.90	0.99	
80 × 80	3.57	1.22	2.58	1.15	2.95	1.32	
60 × 60	3.65	2.03	2.52*	2.21	2.89	2.05	
48 × 48	3.74	3.12	2.39*	3.14	2.12*	3.06	
40 × 40	3.49*	3.94	1.95*	3.98	1.45*	4.10	
HQ	4.28	10.17	3.15	5.05	3.41	5.20	
SQB	2.67	0.23	2.48	0.72	2.87	0.51	
SQB (%)	(+6.8%)	(−8%)	(+0.4%)	(−26.5%)	(+7.5%)	(−19.1%)	

For instance, in Dataset-4, the SQB algorithm achieves an average PSNR gain of 2.87 dB (+7.5% better than 120x120 constant mesh size) with a computation time of 0.51 s (19.1% faster than 120x120 constant mesh size). Comparable results are obtained in Datasets 2 and 3, as shown by the bold lines. Therefore, the proposed SQB adaptive mesh-size method can achieve 5% increase in PSNR quality gain together with a 19% decrease in computation time on the average.

Even if a user could choose an optimal constant mesh size for the entire video sequence, the overall performance of the SQB algorithm cannot be achieved. In practical applications, such a priori information for selecting an ideal mesh size may not be available. The advantage of the proposed SQB algorithm is that it does not rely on any such a priori information; instead, it calculates the mesh-size adaptively based on the blur variance in each frame, which can be calculated in real-time. Overall, the SQB algorithm consistently outperforms the methods with constant mesh-size on average across the entire video sequence.

The differences in the results across the three datasets are due to the distinct blur characteristics present in their frames. For example, Dataset-2 contains more uniformly blurred frames with lower variance compared to the others. Dataset-3 exhibits the highest variance and average blur magnitude. Dataset-4 falls in between the two with moderate blur magnitude. It can be concluded that the proposed approach exceeds the performance of both full-frame and constant mesh-size deblurring methods.

Note that the mesh sizes marked with an asterisk (*) in Table 1 indicate that The specific mesh size was unable to deblur certain frames because the mesh size was smaller than the blur magnitude Consequently, the smaller mesh sizes failed to improve PSNR performance contrary to expectation.

From Table 1, it is observed that a PSNR gain and computation-time trade-off exist. Figure 14 illustrates the computation time versus average PSNR gain for all datasets, comparing constant mesh sizes with the SQB-determined adaptive mesh size. The green circle shows the ideal region where the highest PSNR gain is achieved with minimal computation time. Although this ideal corner is not attainable, results show that the proposed SQB algorithm marginally outperforms any constant-mesh size case, in particular for dataset 3 and 4. Furthermore, observing the result of Dataset-1 in particular, we recognize that further improvement can be made for achieving an optimal trade-off between PSNR gain and computation time. This can be accomplished by further modifying the proposed SQB algorithm to introduce a user selectable trade-off parameter.

Figure 14 Plots for the average computation times and PSNR gains for both SQB and constant mesh-size methods across three datasets as listed in Table 1.

The tested constant mesh sizes include (40, 48, 60, 80, 96, 120, 160, 240, 480). The SQB results (*) consistently outperform the constant mesh-size curves in all datasets, signifying improved performance. The optimal area, marked by the green circle, represents where the highest PSNR gain is achieved with the least computation time.

Introducing a trade-off control parameter for the SQB algorithm

The SQB algorithm was developed by manually labeling optimal mesh sizes for speed and quality within a specific dataset. However, this balanced mesh-size framework might not be ideal for every dataset. Additionally, in order to achieve specific performance goals, users might want to have a control on the algorithm’s selection of the mesh sizes. To address this, we enhanced the SQB method by introducing a tradeoff parameter (λ), creating the parameterized speed-quality balanced (PSQB) mesh size. This is represented in Eq. (14) as follows: (14) mPSQBσs2=λ∗mSQBσs2.

Scaling the SQB mesh size by λ vertically shifts the fitted curve which is shown in Fig. 11.

The effect of tradeoff parameter (λ) is tested on our datasets by varying from 0.1 to 4 with the step of 0.1. The obtained PSNR gains and the corresponding computation times for the constant mesh-size versus the PSQB algorithm for various tradeoff parameter λ is plotted in Fig. 15 together with the SQB points. Examining the obtained graphs, it is evident that the PSQB mesh size results consistently form a curve that is above the constant mesh size counterpart for all selected values of λ. This indicates that the proposed PSQB method consistently outperforms the constant mesh-size method for all λ values. Furthermore, the λ parameter gives the ability to the user to select the desired trade-off between computation time and PSNR according to the application’s needs.

Figure 15 The average PSNR gains versus computation times for constant mesh-size, SQB and PSQB algorithms for the three considered datasets.

Constant mesh-sizes are (40, 48, 60, 80, 96, 120, 160, 240, 480). The trade-off parameter λ is adjusted within the range of 0.1 to 4 yielding the curves for PSQB. In all three datasets, these curves remain consistently above the curves for constant mesh-size, indicating superior performance. The blue point in each curve correspond to the degenerate λ = 1 case (the SQB algorithm).

The PSQB curve is limited with HQ mesh-size. The saturation at the top of the curves in Fig. 15 suggests that the blur magnitude always imposes a lower bound on the mesh size, which none of the adaptive algorithms can reduce further. Thus, with a small tradeoff parameter, the PSQB method converges to the HQ mesh size.

Conclusion

In this study, we proposed an adaptive mesh-size based non-blind video motion deblurring approach aided with inertial measurement sensor. This method segments images into grid cells and estimates local PSFs for each cell. This method is the first to adaptively adjust the grid cell size based on the motion blur characteristics within the image. We also adaptively change overlap region and edge-tapering magnitude according to the motion blur magnitude for reducing ringing and grid cell artifacts. A variance parameter is introduced in order to represent non-uniformity level of the motion blur within the image frame. Two adaptive mesh-size algorithms can be selected, either the SQB (speed-quality balanced) or the HQ (high-quality) algorithm. The HQ algorithm prioritizes achieving the highest quality results regardless of the computation time. In contrast, an optimal mesh size is determined by the SQB method dynamically to obtain a near optimal quality versus computation time point.

The speed quality tradeoff of SQB algorithm is controlled with a parameter λ, so that the user can adjust this parameter according to the required quality and speed. The results demonstrate a significant improvement in both visual quality and computation time in video deblurring relative to the conventional full-frame based and previous constant mesh based deblurring methods on average.

The proposed method also has a computationally parallelizable structure. The mesh-grid structure enables the deconvolution process to be performed independently for each grid cell, ensuring that the computation of one cell does not affect others. This feature supports parallel processing using GPU or FPGA devices, which can significantly boost the speed of the deblurring process. Thus, the computation of this method can be accelerated on appropriate computational hardware for real-time applications.

While the proposed adaptive mesh-size based non-blind video motion deblurring method shows promising results, several limitations should be acknowledged. First, the algorithm relies on inertial measurements for the PSF estimation process, and the performance is directly influenced by the accuracy of the IMU sensor. In our study, the effect of noise or other errors in IMU readings are not considered, which can degrade the deblurring performance. Actually, this drawback exists for all the inertially-aided deblurring techniques which makes quality and accurate sensor calibration crucial for achieving optimal results. Our proposed method may be affected more from the IMU accuracy since the adaptive mesh-size determination process also depends on the IMU data. Additionally, the method assumes that there is a large distance between the scene and the camera, an assumption that may not hold in close-range scenarios. In such cases, the proposed method’s performance could deteriorate, as the motion blur characteristics can be dominated by the translational motion of the camera. Future research could focus on relaxing the assumption of negligible translational motion, enabling the method to be applied effectively in these situations.

Additional Information and Declarations

Competing Interests

Author Contributions

Data Availability

We declare that there are no competing interests.

Ahmet Arslan conceived and designed the experiments, performed the experiments, analyzed the data, performed the computation work, prepared figures and/or tables, authored or reviewed drafts of the article, and approved the final draft.

Gokhan Koray Gultekin conceived and designed the experiments, analyzed the data, performed the computation work, prepared figures and/or tables, authored or reviewed drafts of the article, and approved the final draft.

Afsar Saranli conceived and designed the experiments, analyzed the data, authored or reviewed drafts of the article, and approved the final draft.

The following information was supplied regarding data availability:

The data is available at Zenodo: Gultekin, G. K., Arslan, A., & Saranli, A. (2024). gkgultekin/adaptive-mesh-deblurring: v1.0.3 (v1.0.3). Zenodo. https://doi.org/10.5281/zenodo.13889432.

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
