# Peer review of "IMU-aided adaptive mesh-grid based video motion deblurring"

_PeerJ Computer Science, doi:10.7717/peerj-cs.2540_

## Round 0.1 · original submission · Minor Revisions

The reviewers have suggested several important changes to your manuscript. Please carefully address all these recommendations and resubmit the revised version for further considerations.

*Please note this decision email is only being sent again to fix an error in the system that is preventing you from resubmitting.

Reviewer 1 ·

Basic reporting

This study introduces a novel algorithm for motion deblurring that employs an adaptive mesh-grid technique to address non-uniform motion blur while emphasising the reduction of computational expenses. The suggested approach segments the image into a mesh grid and assesses the blur point spread function utilising an inertial sensor.The topic is interesting. Some comments need to be addressed before acceptance


In the abstract, could you please define PSF?
In the abstract, please add numerical findings?


In the introduction, please add a reference to the first paragraph.

It is better to highlight the novelty and contributions in points.

In the related work, please remove the full name Point Spread Function” (PSF) as you already defined it in the introduction.

Please summarize the literature in a Table.

Experimental design

What is the mechanism by which the mesh-grid decomposition process partitions the image frame into grid cells?

How is the area where grid cells overlap identified?

Validity of the findings

Please define performance metrics.


Are there any benchmark datasets available to test your algorithm?

Figure 12 resolution is poor.


Please add the limitations of this study and future work.

Cite this review as

Reviewer 2 ·

Basic reporting

Raw data sets not shared

Experimental design

no comment

Validity of the findings

no comment

Additional comments

Thank you for submitting your work to PeerJ computer science.

This paper tackles the interesting problem of motion deblurring. The proposed approach adaptively divides the image into a mesh grid and uses an inertial sensor to determine the blur PSF.

My main concern with the existing manuscript is the absence of the raw data. I wasn't able to find the datasets used by the experimental evaluation as part of the submitted material. I would like to encourage the authors to make the dataset available and then resubmit the article.

Cite this review as

---

## Round 0.2 · accepted · Accept

Congratulations, the reviewers are satisfied with the revision and recommended accept decision.

Reviewer 1 ·

Basic reporting

I would like to thank the authors for addressing all my comments

Experimental design

The authors have addressed my comments

Validity of the findings

The authors have addressed my comments

Cite this review as

Reviewer 2 ·

Basic reporting

no comment

Experimental design

no comment

Validity of the findings

no comment

Additional comments

Raw data provided as requested.

Cite this review as